# Prospective randomised trial examining the impact of an educational intervention versus usual care on anticoagulation therapy control based on an SAMe-TT$_2$R$_2$ score-guided strategy in anticoagulant-naïve Thai patients with atrial fibrillation (TREATS-AF): a study protocol

Arintaya Phrommintikul [1,2] Surakit Nathisuwan,[3] Siriluck Gunaparn,[1] Rungroj Krittayaphong,[4] Wanwarang Wongcharoen,[1] Sukhi Sehmi,[5] Samir Mehta,[5] Neil Winkles,[5] Peter Brocklehurst,[5] Jonathan Mathers [6] Sue Jowett,[7] Kate Jolly,[6] Deirdre Lane [8] G Neil Thomas [9] Gregory Y H Lip [10,11] TREATS-AF Study Group

GNT and GYHL are joint senior authors.

For numbered affiliations see end of article.

**Correspondence to**
Professor Gregory Y H Lip;
Gregory.Lip@liverpool.ac.uk

## ABSTRACT

**Introduction** The burden of atrial fibrillation (AF) in Thailand is high and associated with increased morbidity, mortality and healthcare costs. Vitamin K antagonists (eg, warfarin), commonly used for stroke prevention in patients with AF in Thailand, are effective but are often suboptimally controlled. We aim to evaluate the impact of an SAMe-TT$_2$R$_2$ score-guided strategy and educational intervention compared to usual care on anticoagulation control expressed by the time in therapeutic range (TTR) at 12 months, in anticoagulant-naïve Thai patients with AF.
**Methods and analysis** Multicentre, open-label, parallel-group, randomised controlled trial conducted in Thailand among adult patients (age: 18 years) with AF who are anticoagulant naïve. Patients will be randomised to one of two groups; an SAMe-TT$_2$R$_2$ score-guided strategy with educational intervention and usual care versus usual care alone. The planned follow-up period is 12 months. The primary outcome is TTR at 12 months. Secondary outcomes include: (1) TTR at 6 months; (2) thromboembolic and bleeding events at 12 months; (3) composite major adverse cardiovascular events at 12 months; (4) change in patients' knowledge of AF between baseline and 6 months and 12 months; (5) cost effectiveness; (6) quality of life at baseline, 6 months and 12 months using EQ-5D-5L (Thai version) and (7) patient satisfaction/perceptions of the TREAT intervention. An embedded qualitative study will assess patient perceptions of the TREAT intervention.
**Ethics and dissemination** The study has been approved by the Ethical Review Committee, Ministry of Public Health of Thailand, and registered in the Thai Clinical Trials Registry. The results of this trial will be submitted for publication in a peer-reviewed journal. Participants will be informed via a link to a preview of the publication. A lay

## Strengths and limitations of this study

► Warfarin remains the default therapy for stroke prevention in atrial fibrillation (AF) in many countries, but is often suboptimally managed.
► A one-time structured educational-behavioural intervention (TREAT intervention) among those predicted to be less likely to achieve optimal control may be a simple and cost-effective adjunct management strategy, which could help improve time in therapeutic range.
► This prospective individually randomised controlled trial in anticoagulant-naïve patients with AF aims to assess whether the application of an SAMe-TT$_2$R$_2$ score-guided strategy and TREAT intervention plus usual care could improve anticoagulation control with warfarin.
► The study design has a multidisciplinary team approach, and the efficacy, safety, patients' satisfaction as well as cost effectiveness will be assessed.
► One challenge of the study, if implemented, is to standardise the quality of usual care and TREAT intervention among centres.

summary will also be provided to all participants prior to publication.
**Trial registration number** TCTR20180711003.

## INTRODUCTION

Atrial fibrillation (AF) is associated with decreased quality of life, and increased mortality and morbidity from stroke/thromboembolism.[1][2] Incidence of AF has been

increasing in developing countries, including Thailand, partly due to the rapid change of population dynamics towards an ageing society.[3 4] The hospitalisation rate for Thai patients with AF was high (15.5 per 100 000 person-years), with a high mortality (44.0% after 72-month follow-up (average 46 months)).[5] The high burden of AF in Thailand is also associated with similarly high health-care costs.[6]

Appropriate oral anticoagulation, either with a vitamin K antagonist (VKA, eg, warfarin) or with a non-VKA oral anticoagulants (NOAC), is essential for effective stroke prevention in patients with AF with one or more stroke risk factors.[7] Despite the advantages of NOAC over VKA, the access of NOAC are generally limited due to cost. Hence, VKAs are widely used globally.[8 9] In Thailand, warfarin is the default therapy for stroke prevention in AF but is often suboptimally managed due to the many inherent limitations associated with this drug, including diet, drug and alcohol interactions and difficulties with anticoagulation monitoring. Recent data from retrospective cohort studies, along with a nationwide registry, suggested that the average time in therapeutic range (TTR) among patients with AF in Thailand ranged from 50% to 55%.[10–13] Those with suboptimal anticoagulation control, TTR <65%, had a twofold–threefold increased risk of stroke, major bleeding and death compared with those with TTR ≥65%.[12] As a result, strategies to improve TTR control are clearly needed and would likely have a major impact on preventing these adverse events.

TTR is influenced by numerous modifiable and non-modifiable factors, but the more common clinical factors have been used to formulate the SAMe-TT$_2$R$_2$ score (table 1).[14]

Several observational studies including those conducted in Asian populations, have demonstrated that an SAMe-TT$_2$R$_2$ value of >2 is predictive of poor TTR, all-cause mortality and the composite endpoint of thromboembolic events, major bleeding and mortality.[13 15–17] By identifying those who are at risk of poor TTR, targeted efforts to improve TTR can be effectively provided to the vulnerable group. If TTR can be improved to >65%, VKAs can provide comparable efficacy and safety to that seen with NOACs.[18 19] Previo studies have demonstrated that many patients with AF possess little knowledge about AF and do not understand the risks/benefits of OAC,[20–25] and this may contribute to poor international normalised ratio (INR) control.

We have previously shown that an educational-behavioural intervention (TREAT, ISRCTN93952605) significantly improved TTR 6 months after warfarin initiation compared with usual care alone (78.5% vs 66.7%, respectively; p=0.01).[26–28] Thus, a one-time structured educational-behavioural intervention among those predicted to be less likely to achieve good INR control may be a simple and cost-effective adjunct management strategy, which could help improve individual TTR. Increased patient understanding of disease and treatment, and a reduction of adverse events, may improve quality of life.

We, therefore, conducted a prospective individually randomised controlled trial (RCT) in anticoagulant-naïve Thai patients with AF to assess whether the application of an SAMe-TT$_2$R$_2$ score-guided strategy and TREAT intervention[26–28] plus usual care could improve anticoagulation control with warfarin (measured by percentage TTR) at 12 months (primary outcome) compared with usual care alone.

## Objectives

The primary objective of this study is to evaluate the impact of an SAMe-TT$_2$R$_2$ score-guided strategy and educational intervention (plus usual care) compared with usual care alone, on patient's anticoagulation control with warfarin, as measured by the TTR at 12 months, in anticoagulant-naïve Thai patients with AF. As secondary objectives, we aim to evaluate the impact of this intervention on patient knowledge, TTR at 6 months, thromboembolic and bleeding events and the composite major adverse cardiovascular events (MACE). We will also perform analysis to evaluate the cost effectiveness of the intervention and investigate patients' satisfaction and acceptance of the TREAT intervention using a brief questionnaire and further in a qualitative sub-study.

**Table 1** Components of the SAMe-TT$_2$R$_2$ score

| Components | | Points |
|---|---|---|
| S | Sex (female) | 1 |
| A | Age (<60 years) | 1 |
| Me | Medical history* | 1 |
| T | Treatment (interacting drug, eg, amiodarone) | 1 |
| T | Tobacco use (within 2 years) | 2 |
| R | Race (non-Caucasian) | 2 |
| Maximum total | | 8 |

SAMe-TT$_2$R$_2$ scores 0–2: the patients with AF likely to achieve and maintain optimal TTR, and SAMe-TT$_2$R$_2$ scores >2: the patients with AF likely poorer responders.
*More than two of the following: hypertension, diabetes mellitus, coronary artery disease/myocardial infarction, peripheral atrial disease, congestive heart failure, previous stroke, pulmonary disease and hepatic or renal disease.
AF, atrial fibrillation; TTR, time in therapeutic range.

## METHODS
### Study design and setting

This is a multicentre, open-label, paralleled-group, RCT conducted among adult Thai patients (age: ≥18 years) with AF who are anticoagulant naïve. Patients who are eligible for stroke prevention according to the ESC Guidelines for the diagnosis and management of AF[1] (men with CHA$_2$DS$_2$VASc score ≥1; women with CHA$_2$DS$_2$VASc score ≥2) will be randomised to one of two groups, either an SAMe-TT$_2$R$_2$ score-guided strategy (see table 1)

and educational intervention plus usual care versus usual care alone. The planned follow-up period is 12 months. Study sites are seven public hospitals across six provinces (Bangkok, Chiang Mai, Chiang Rai, Lampang, Khon Kaen and Nakhon Ratchasima) in Thailand. To promote generalisability of our findings, these seven hospitals are a combination of three university hospitals (Maharaj Nakorn Chiang Mai Hospital, Siriraj Hospital, and Queen Sirikit Heart Centre of the Northeast) and four secondary/tertiary care hospitals (Maharat Nakhon Ratchasima Hospital, Chiang Rai Prachanukroh Hospital, Nakornping Hospital and Lampang Hospital). The study sites and investigators are listed in online supplemental appendix 1. The study protocol was approved by the Institutional Ethical Committee of each study site and by the Institutional Review Board (IRB) of the Ministry of Public Health, the Royal Government of Thailand (Central Research Committee (CREC) number: COA-CREC 007/2020) and registered with the Thai Clinical Trials Registry.

## Participants

Newly diagnosed adult patients with AF identified and referred from cardiology, internal medicine, family medicine and general clinics at the hospitals and referrals from primary care will be eligible for enrolment into the study.

## Eligibility criteria

Patients must meet all of the following inclusion criteria to be eligible for randomisation:
1. ≥18 years of age
2. Newly diagnosed patients with non-valvular AF.
3. ECG-documented AF
4. Warfarin eligible (men with $CHA_2DS_2VASc$ score ≥1; women with $CHA_2DS_2VASc$ score ≥2)
5. Warfarin naïve (no treatment with anticoagulation within the past 12 months; treatment may have started within the prior 28 days from randomisation).
6. Able to comply with scheduled visits, treatment plan and laboratory tests
7. Able to give informed consent and comply with study protocol (with support of a carer)

Patients will be excluded from participation if they present with any of the following:
1. Any contraindication to oral anticoagulation
2. Prosthetic cardiac valve or significant valvular heart disease with an indication for heart surgery
3. Likelihood of intermittent or permanent discontinuation of warfarin during follow-up (eg, major surgery or post-AF ablation)
4. Known active malignancy with a life expectancy less than 5 years.
5. Diagnosed significant cognitive impairment preventing provision of informed consent and/or able to comply with the study protocol.
6. Any disease likely to cause death within 12 months.

The study was originally designed to enrol the patients within 5 days of warfarin treatment; however, the average

referral time was 14 days. The trial steering committee agreed to extend the duration between starting warfarin and randomisation to 28 days to improve recruitment rate, external validity and generalisability of the trials. The amended study protocol was approved by the Institutional Ethical Committee of each study site and by the IRB of the Ministry of Public Health, the Royal Government of Thailand and updated in the Thai Clinical Trials Registry.

## Randomisation

After participant eligibility has been confirmed and informed consent has been provided, the participant will be randomised into the trial. Randomisation will be done using a web-based platform with blinded allocation. Randomisation will be stratified based on centre, sex (male or female) and baseline $SAMe-TT_2R_2$ score (0–2, 3–5 and 6–8). Participants will be randomised at the level of the individual in a 1:1 ratio to either the $SAMe-TT_2R_2$ score-guided strategy and educational intervention plus usual care (intervention group) or usual care alone (figures 1 and 2). Group 1 is the usual care group using warfarin (control group). For Group 2, the patients are further divided into two groups. Group 2a are patients with an $SAMe-TT_2R_2$ score of 0–2 who will receive warfarin plus usual care. Group 2b are patients with an $SAMe-TT_2R_2$ score of >2 who will receive warfarin plus the TREAT education-behavioural intervention as an adjunct to their regular INR monitoring, to improve their TTR.

## Trial intervention

As described previously, the TREAT intervention is a patient-centred intervention for patients with AF co-developed through the integration of theoretical and clinical frameworks, and patient feedback.[26 27] This behaviour-change intervention package consists of an educational booklet, diary, worksheet and a DVD for reinforcement. The educational booklet covers AF causes and consequences, warfarin and its metabolism, stroke risk and risk of bleeding on treatment and lifestyle changes (diet, alcohol and lifestyle change). The patient DVD contains health professionals and 'expert patient' narratives that covers AF causes, consequences, side effects, treatment options, warfarin, INR monitoring, lifestyle changes, along with common psychological and physical barriers to anticoagulation that patients may experience. The patient worksheets include a simple self-calculation of stroke risk ($CHA_2DS_2-VASc$ score), personal barriers to uptake of warfarin and discussion of personal goals for lifestyle changes. In addition, a self-monitoring diary, including diet, alcohol intake (in units), warfarin regimen and INR results is also provided.

To overcome the language barrier of the general Thai public toward English, the TREAT intervention was culturally adapted to fit the local context and then translated. Forward and backward translation by two bilingual experts was performed. A test run of the Thai version of the TREAT intervention was conducted among patients

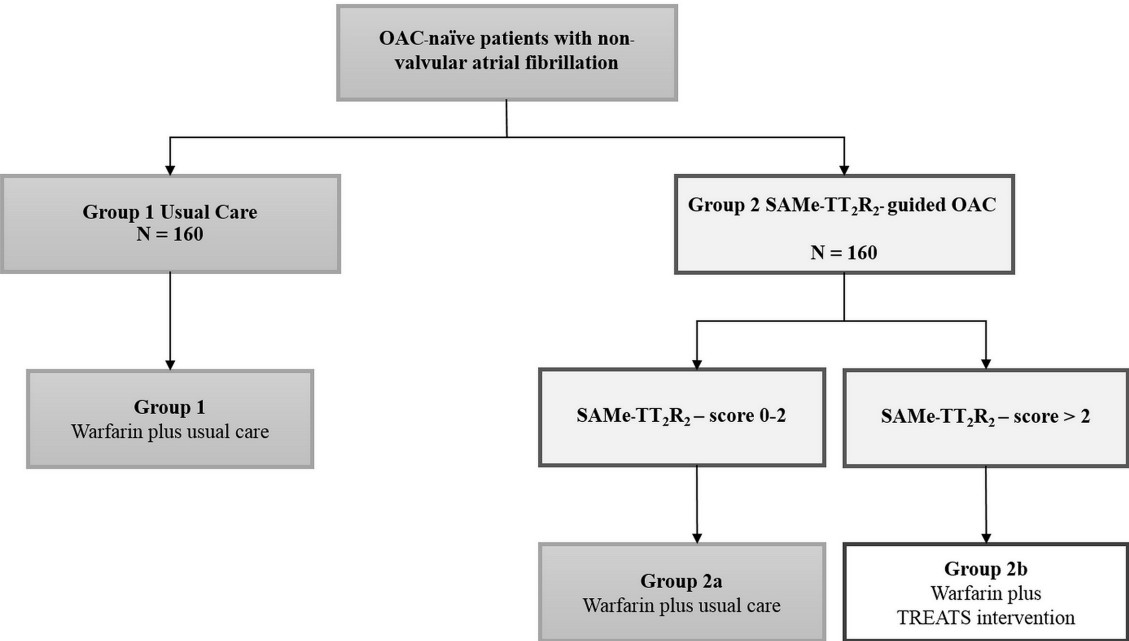

**Figure 1** Study design and recruitment. Group 1: usual care using warfarin (control). Group 2: care pathway designated based on stratification of patients using the SAMe-TT$_2$R$_2$ score (warfarin plus usual care). Group 2 care pathways: Group 2a: SAMe-TT$_2$R$_2$ score 0–2 (warfarin plus usual care). NB: usual care would include management in the anticoagulation clinic, and patient information about AF and need for warfarin by a healthcare professional using the standard warfarin education checklist. Group 2b: SAMe-TT$_2$R$_2$ score >2 (warfarin plus intensive TREAT education-behavioural intervention as an adjunct to their regular INR monitoring to improve their TTR on warfarin). AF, atrial fibrillation; OAC, oral anticoagulants; TTR, time in therapeutic range.

with AF with a range of educational ability (Thai literacy level is 94%) and age levels. Patient feedback was incorporated to improve the usability of the Thai-version TREAT intervention.

A training session was conducted to prepare all study site coordinators (pharmacists and nurses) on how to deliver the intervention effectively. To avoid contamination, those who provide the TREAT intervention and those who provide usual care will be two different groups of pharmacists and nurses. The intervention will be delivered within 4 weeks of initiating warfarin. Intervention fidelity will be ensured by direct observation of random sessions at each centre by the trainer. Patients in the usual care group will receive training on the OAC management

from the anticoagulation clinic, and patient information about AF and need for warfarin by a healthcare professional using the standard warfarin education checklist.

### Patient and public involvement

Patients or the public were not involved in the design, or conduct, or reporting, or dissemination plans of our research. The patients involved in the development of Thai version of the TREAT intervention.

### Outcome measures

The primary outcome is TTR at 12 months. Secondary outcomes include (1) TTR at 6 months; (2) thromboembolic and bleeding events from at 12 months; (3) composite

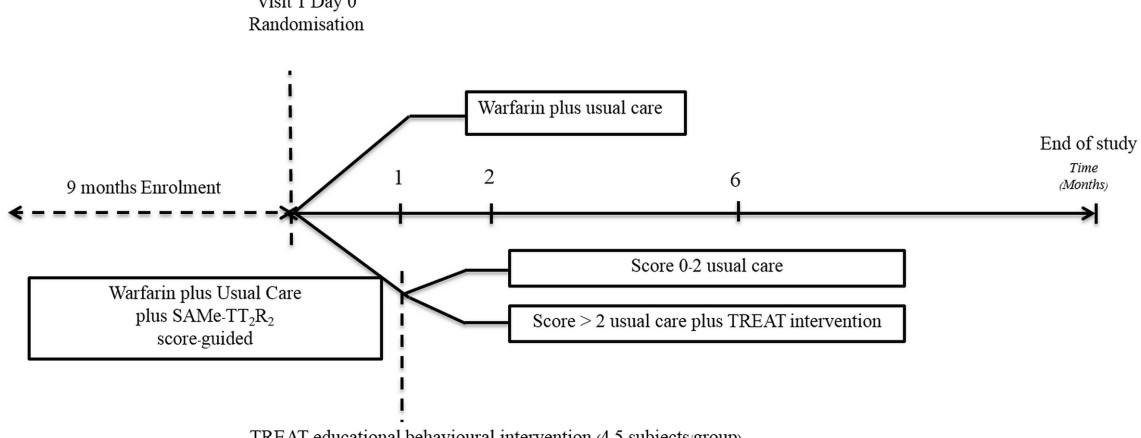

**Figure 2** Timing of randomisation, study visits and follow-up.

**Table 2** Study visits schedule

**Clinical assessment**
a.  Demographic information: age, sex, ethnicity, education level, height (cm), weight (kg), body mass index, smoking status and alcohol use
b.  Medical history: history of arterial hypertension, heart failure, diabetes mellitus, coronary artery disease (myocardial infarction, PCI and CABG), peripheral artery disease, stroke, transient ischaemic attack, systemic embolism, COPD, chronic kidney disease, chronic liver disease, thyroid disease and bleeding event
c.  Physical examination, blood pressure, heart rate, EHRA and NYHA class
d.  Concomitant medical therapy

**OAC therapy education:** all patients—a standard OAC education session conducted by a healthcare professional about the clinical significance and management of OAC therapy
**Questionnaires:** atrial fibrillation knowledge questionnaires, patient satisfaction (Likert scale) and EQ-5D-5L (Thai version)
**Health economics assessment:** EQ-5D-5L (Thai version), AF, warfarin and clinical event-related healthcare resource use and intervention costs

**Follow-up of clinical events:** TTR, composite of thromboembolism, major bleeding and all-cause death (in all patients)

|  | **Baseline** | **M1±7D** | **M2±7D** | **M6±14D** | **M12±14D** |
|---|---|---|---|---|---|
| Clinical assessment | **a,b,c,d** | **c** | **c** | **c** | **c,d** |
| ECG | x |  |  |  |  |
| INR** | x | x | x | x | x |
| Standard OAC therapy education | x |  |  |  |  |
| TREAT intervention |  | x |  |  |  |
| Questionnaires | x |  |  | x | x |
| Qualitative assessment | x† |  |  | x |  |
| Health economic assessment | x |  |  | x | x |
| Follow-up of clinical events |  |  |  | x | x |

*INR monitoring is mandatory during the 12-month follow-up period for all patients, with frequency as per usual care conducted by anticoagulation clinics.
†Baseline interviews of patients will take place within 4 weeks of receiving the intervention.
AF, atrial fibrillation; CABG, coronary artery bypass grafting; COPD, chronic obstructive pulmonary disease; ECG, electrocardiogram; EQ-5D-5L, The 5-level EQ-5D version; INR, international normalised ratio; OAC, oral anticoagulants; PCI, percutaneous coronary intervention.

MACE outcome of non-fatal MI, non-fatal stroke and cardiovascular death and all-cause mortality, in an exploratory analysis (combined and individually) at 12 months; (4) change in patients' knowledge of AF from baseline to 6 months and 12 months; (5) cost effectiveness; (6) quality of life at baseline, 6 months and 12 months assessed using the 5-level EQ-5D version (EQ-5D-5L) (Thai version)[29 30] and (7) patient satisfaction/perceptions of the TREAT intervention. An embedded qualitative study will assess patient perceptions of the TREAT intervention.

### Study procedures and schedules of assessment
Baseline SAMe-$TT_2R_2$ score will be calculated (table 1). All patients will receive warfarin, dose adjusted to achieve a target INR of 2.0–3.0. INR monitoring will be undertaken at routine intervals (as would occur in usual care) and will be recorded over a 12-month period. Patients will be randomised as described in figure 1, either to Group 1 (usual care) or Group 2 (intervention arm), for which the trial is powered. Group 2 is a SAMe-$TT_2R_2$ scores directed management group based on a: patients with AF likelihood of achieving and maintaining optimal TTR (SAMe-$TT_2R_2$ scores 0–2; Group 2a) and those likely to achieve poor TTR (SAMe-$TT_2R_2$ scores>2, Group 2b). From our prior data, approximately 15%–20% will likely enter Group 2a. The primary and secondary outcomes between Group 1 and Group 2 (together) will be compared. The schedule of assessment is presented in table 2. INR monitoring is independent of the intervention (ie, naturalistic) as we recognise that the improvement of TTR may be a

result of more frequent monitoring. Patients in all groups will have their INR monitored as they would if they were not in the trial (by anticoagulation services which are separate from the trial and who are blinded to treatment allocation).

### Data collection and management
Medical records of the participating hospitals serve as the source data, including clinical information and laboratory values, which are accessible and maintained in accordance to national standards and regulations. Outcome data will be extracted from participant's clinical notes and laboratory reports into an electronic case record form (CRF). For long-term storage, clinical data will be electronically archived at Faculty of Medicine, Chiang Mai University, Thailand, as per the regulatory requirements for good clinical practice and in line with the Medical Research Council requirements and local policies.

The questionnaires will be completed by the patient. For those with literacy problems, the questionnaires will be read out to the participant, the participant will respond to each question and their responses will be recorded by the staff member administering the questionnaires.

Trial data will be captured in a Research Electronic Data Capture (REDCap), secure SQL server database. An independent data monitoring committee (DMC), including both local and international experts, has been appointed and convened. Interim analyses of major outcome measures and safety data will be conducted and provided in strict

confidence to the DMC. Any decision to stop the trial early will be based on the balance of efficacy and safety.

## Sample size and power calculations

Most recent studies suggest that the mean TTR in Thai patients receiving warfarin is in the range of 50%–55%.[11 12] Since an improvement of 10% in the mean TTR is likely to be clinically meaningful, we assume a 10% difference in 12-month TTR with 90% power and 5% significance level for our sample size calculation and using an SD of '26' as observed in a previous Thai study to determine the sample size estimates. Hence, to detect a 10% mean difference in TTR at 12 months, using two-sided test, sample size for randomisation in a 1:1 ratio would be 288 or 144 per arm. Assuming and adjusting for a 10% attrition/loss to follow-up rate, a total sample size of 320 patients or 160 per arm is needed.

## Statistical analysis

Continuous variables will be presented as mean (SD), or median (IQR), as appropriate; categorical variables will be reported as counts with percentages. Descriptive statistics will be presented for baseline demographic and clinical information. For all major outcomes, summary statistics and differences between groups (eg, mean differences and relative risks) will be presented, with CIs and p values from two-sided tests also given.

All analyses will be based on the intention-to-treat (ITT) principle. The primary comparison groups will be composed of those randomised to Group 1, usual care, versus those randomised to Group 2, the SAMe-TT$_2$R$_2$ score-guided strategy with the TREAT intervention (plus usual care). All outcomes will be adjusted for the stratification variables (centre, sex and baseline SAMe-TT$_2$R$_2$ score) where possible. Two-tailed p values <0.05 will be considered statistically significant. No adjustment for multiple comparisons will be made.

## Subgroup analysis

Subgroup analyses will be conducted for the stratification variable sex only. All other stratification variables will be used only to ensure that the two groups are balanced in terms of these important variables. Subgroup analyses will be limited to the primary outcome only. Tests for statistical heterogeneity (eg, by including the treatment group by subgroup interaction parameter in the regression model) will be performed prior to any examination of effect estimates within subgroups. The results of subgroup analyses will be treated with caution and will be used for the purposes of hypothesis generation only.

## Missing data and sensitivity analyses

Missing and ambiguous data will be queried using a data clarification system, and will focus on data required for trial outcome analysis and safety reporting. Every attempt will be made to collect full follow-up data on all study participants. A sensitivity analysis will be performed on the primary outcome measure only to examine the possible impact of any missing data for the primary outcome. This will include the use of multiple imputation using chained equations. Further sensitivity analysis will also be done if a substantial number of participants do not comply with randomised allocated group and so a per-protocol analysis may also be conducted. Any sensitivity analyses will not, irrespective of their differences, supplant the planned primary analyses.

## Interim analysis

Interim analyses of safety and efficacy for presentation to the independent DMC will take place during the study. The committee will meet prior to study commencement to agree the manner and timing of such analyses. The analysis of the primary and major secondary outcomes and full assessment of safety (adverse events and serious adverse events) will be performed at annual intervals. Criteria for stopping or modifying the study based on this information will be ratified by the DMC.

## Planned final analyses

The final analysis for the study will occur once the last randomised participant has completed the 12-month follow-up and any corresponding outcome data have been entered onto the study database and validated as being ready for analysis.

## Withdrawal

Patients may withdraw consent from the study at any time. Patients may also withdraw from trial treatment but continue with study follow-up and data collection as per the protocol. If the withdrawal is initiated by a healthcare professional, full details for the reason for withdrawal will be recorded on the case report forms. In all other cases, a simple statement reflecting the patient's preference will be noted. The patients who are switched to NOACs will be followed for clinical events. The INR data will be collected to the end of warfarin treatment for the TTR calculation and analysis as ITT analysis.

## Analysis of outcome measures
### Primary outcome

The primary outcome is the TTR at 12 months and differences in TTR between the intervention and usual care groups will be examined using a linear regression model. Results will be presented as mean difference and 95% CI. A key assumption for this analysis is that the data will be normally distributed; this assumption will be examined visually based on the residuals from the fitted model. If data are deemed not to be normally distributed, then the Mann-Whitney U test will be used for analysis.

### Secondary outcomes

The secondary endpoints for the trial includes continuous, categorical and time-to-event data items. Any secondary endpoints that are continuous in nature will be analysed in the same way as described for the primary outcome. For dichotomous secondary endpoints, the proportion of participants and percentages will be compared between arms using a log-binomial model.

Relative risks and 95% CIs will be calculated. For any time-to-event outcomes these will be compared between treatment arms by using survival analysis methods. Cox proportional hazards model will be used and treatment effects will be expressed as HRs with 95% CIs. Kaplan-Meier survival curves will be constructed for visual presentation of time-to-event comparisons.

## Qualitative study

Qualitative data will be obtained directly from patient interviews and interviews with healthcare professionals delivering the intervention as part of an embedded qualitative substudy. A purposive sample of 10–15 participants from the intervention arm will take part in one-to-one semi-structured interviews with a qualitative researcher, at two time-points (within 4 weeks of the intervention and at 6-months follow-up). Interviews will explore participants' experience of AF; of receiving the TREAT intervention; their perspectives on the utility of different interventions components; as well as behavioural modifications resulting from the intervention. Staff delivering the intervention at each study site will also be interviewed to understand their perceptions of the TREAT intervention and their experience of delivering it to patients. A thematic analysis of interview content will be informed by the framework analytic approach. Audio-recorded interviews will be translated and transcribed into English in order to facilitate team-based analysis by Thai and English research team members.[31]

## Health economics analysis

The economic evaluation will determine the cost effectiveness of the intensive educational intervention versus usual care. The evaluation will take the form of an incremental cost–utility analysis to estimate cost per quality adjusted life year (QALY) over 12-months follow-up from a Thailand health service perspective. If differences in TTR are achieved in the trial, a longer-term Markov model-based cost per QALY analysis will be also undertaken.

### Data collection

Data will be collected on healthcare utilisation required for the intervention, warfarin therapy and monitoring, and healthcare visits, hospitalisations, investigations and medication related to thromboembolism, bleeding and management of AF. Information will be collected at the 6-month and 12-month patient visits from trial CRFs and hospital records. Additional costings, such as those associated with transportation to the hospital, will be collected from the patients. The EQ-5D-5L (Thai version) questionnaire will be administered to participants at baseline, 6 months and 12 months to estimate QALYs.

### Data analysis

Unit costs from standard Thai sources will be applied to all healthcare resource use items, and mean resource use (for each category of healthcare usage) and mean total costs will be calculated for both trial arms. The current Thailand's value set will be applied to patient EQ-5D-5L responses to obtain utility scores and QALYs will be calculated using the area under the curve approach. Multiple imputation will be used to impute all missing values for the EQ-5D-5L and total cost estimates for non-responders. As cost and QALY data are likely to have a skewed distribution, a non-parametric comparison of means (using bootstrapping) will be undertaken. A cost–consequence analysis will initially be reported, describing all the important results relating to resource use, costs and consequences. Incremental cost-effectiveness and cost–utility analyses will then be undertaken to estimate the incremental cost per QALY gained at 12 months, with adjustment for baseline covariates. The robustness of the results will be explored using sensitivity analysis. Cost-effectiveness acceptability curves (CEACs) will also be produced to reflect the probability that the intervention will be cost effective at different cost per QALY willingness to pay thresholds.

### Model-based analysis

If the trial data show a benefit in TTR due to the intervention, trial results will be extrapolated using a Markov model-based cost per QALY. This model will take into account the long-term impact of anticoagulant control on thromboembolism and bleeding over the patient's lifetime, with costs and outcomes discounted at 3%. The model structure will be informed by previously published modelling studies and expert opinion. Extensive deterministic sensitivity analysis using predefined ranges will be undertaken to explore which model parameters have most impact on cost effectiveness. Parameter estimates will be incorporated into the model as distributions to allow for a probabilistic sensitivity analysis, with CEACs constructed to assess the probability of cost effectiveness at different cost per QALY willingness to pay thresholds.

## Trial status

The study protocol V.2.1 dated 19 August 2019 was approved by the Institutional Ethical Committee of each study site and by the IRB of the Ministry of Public Health, the Royal Government of Thailand (CREC number: COA-CREC 007/2020) and the study protocol had amended and approved at V.3.0 dated 9 June 2020. The recruitment of participants started in 31 January 2020. Last patient recruitment is expected in May 2021. Data collection is expected to be completed in December 2021.

## Dissemination

Results of this trial will be submitted for publication in a peer-reviewed journal. The manuscript will be prepared by the principal investigators or their delegates and authorship will be determined by the trial publication policy. Participants will be informed of the outcome of the trial via a link to a preview of the publication. A lay summary will also be provided via email or posted to participants prior to publication.

**Author affiliations**
[1]Division of Cardiology, Department of Internal Medicine, Faculty of Medicine, Chiang Mai University, Chiang Mai, Thailand
[2]Center for Medical Excellence, Faculty of Medicine, Chiang Mai, Thailand

[3]Department of Pharmacy, Faculty of Pharmacy, Mahidol University, Bangkok, Thailand

[4]Faculty of Medicine Siriraj Hospital, Mahidol University, Bangkok, Thailand

[5]Clinical Trials Unit, University of Birmingham, Birmingham, UK

[6]Institute of Applied Health Research, University of Birmingham, Birmingham, UK

[7]Health Economics Unit, Institute of Applied Health Research, University of Birmingham, Birmingham, UK

[8]Department of Cardiovascular Health, Faculty of Health and Life Sciences, University of Liverpool, Liverpool, UK

[9]Department of Public Health and Epidemiology, Institute of Applied Health Research, University of Birmingham, Birmingham, UK

[10]Liverpool Centre for Cardiovascular Science, University of Liverpool and Liverpool Heart & Chest Hospital, Liverpool, UK

[11]Department of Clinical Medicine, Aalborg Universitet, Aalborg, Denmark

**Acknowledgements** We acknowledge the assistance and facilities provided by the Medical Research Council (Birmingham Clinical Research Facility).

**Collaborators** TREATS-AF study group: Narawudt Prasertwitayakij, Wattana Wongtheptien, Thanyaluck Chotayaporn, Natrawee Bureekam, Bancha Sookananchai, Vichai Senthong, Antika Wongthanee, Gemma Slinn and Anita Slade.

**Contributors** AP, GNT and GL initiated the study. AP is the principal investigator of RCT. SG is the clinical trial manager. AP, GNT, SN and GL led the protocol development with contribution from SN, SG, RK, WW, SS, SM, NW, PB, JM, SJ, KJ and DL. SN and WW prepared educational content, including educational booklet, diary, worksheet and a DVD. SN conducted TREAT intervention training for pharmacists and monitored during the trial. AP, WW and RK conducted intervention activities together with research assistants. All authors read, contributed to and approved the final version of the manuscript.

**Funding** This study is supported by the Newton Fund through the collaboration of the Medical Research Council, the UK (grant reference number: MR/R020892/1) and the Thailand Research Fund, Thailand (grant reference number: DBG6180009).

**Competing interests** All other authors have no actual or potential conflict of interest capable of influencing judgment on the part of any author on this work. AP and WW: reports speaker fees from Bayer, Pfizer, Boehringer Ingelheim, and Daiichi-Sankyo outside the submitted work. DL: reports investigator-initiated educational grants from Bristol-Myers Squibb (BMS); speaker fees for Boehringer Ingelheim, Bayer and BMS/Pfizer; and consultancy for BMS/Pfizer, Boehringer Ingelheim and Daiichi-Sankyo, all outside of the submitted work. GL: reports consultancy and speaker fees for BMS/Pfizer, Boehringer Ingelheim and Daiichi-Sankyo. No fees are received personally.

**Patient and public involvement** Patients and/or the public were not involved in the design, or conduct, or reporting, or dissemination plans of this research.

**Patient consent for publication** Not applicable.

**Provenance and peer review** Not commissioned; externally peer reviewed.

**Data availability statement** Data are available upon reasonable request.

**ORCID iDs**
Arintaya Phrommintikul http://orcid.org/0000-0003-3986-1951
Jonathan Mathers http://orcid.org/0000-0001-6651-6286
Deirdre Lane http://orcid.org/0000-0002-5604-9378
G Neil Thomas http://orcid.org/0000-0002-2777-1847
Gregory Y H Lip http://orcid.org/0000-0002-7566-1626

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
