## [Reviewer comments · BMJ Open]

ARTICLE DETAILS

TITLE (PROVISIONAL)	Prospective randomised trial examining the impact of an educational intervention versus usual care on anticoagulation therapy control based on a SAME-TT2R2 score guided strategy in anticoagulant-naïve Thai patients with atrial fibrillation (TREATS-AF): study protocol
AUTHORS	Phrommintikul, Arintaya; Nathisuwan, S; Gunaparn, Siriluck; Krittayaphong, Rungroj; Wongcharoen, Wanwarang; Sehmi, Sukhi; Mehta, Samir; Winkles, Neil; Brocklehurst, Peter; Mathers, Jonathan; Jowett, Sue; Jolly, Kate; Lane, Deirdre; Thomas, Neil; Lip, Gregory

VERSION 1 – REVIEW

REVIEWER	Yoshimura, Sohei National Cerebral and Cardiovascular Center
REVIEW RETURNED	28-Apr-2021

GENERAL COMMENTS	Authors aim to evaluate the impact of a SAME-TT2R2 score guided strategy and educational intervention compared to usual care on anticoagulation control expressed by the time in therapeutic range (TTR) at 12-months, in anticoagulant-naïve Thai patients with AF. The manuscript was well written. Minor Comment; # Abstract (Page4/72, line3):Typographical error; (VKA, e.g. warfarin, commonly→ (VKA, e.g. warfarin), commonly
--

REVIEWER	Obamiro, Kehinde University of Tasmania
REVIEW RETURNED	12-Aug-2021

GENERAL COMMENTS	The study is well designed and it will be interesting to see the result. I also like that patient reported outcomes were included in the protocol. The author will do well to provide a justification for the following inclusion criteria: warfarin-eligible (men with CHA2DS2VASc score ≥ 1; women with CHA2DS2VASc score ≥ 2). Of course female gender is a known risk factor for AF-related stroke, the authors can provide a rationale for focusing on women with CHA2DS2VASc score ≥ 2 Are the authors following a specific guideline(AHA/ESC) for stroke prevention? Secondly, the authors will do well to address how patients who have already being prescribed DOACs will be managed. Will they be excluded?
--

	Overall, the protocol is well written and scientifically grounded? Thank you.
--	---

VERSION 1 – AUTHOR RESPONSE

Response to the reviewers

Reviewer 1

Minor Comment;

Abstract (Page4/72, line3):Typographical error; (VKA, e.g. warfarin, commonly→ (VKA, e.g. warfarin), commonly

Response: We thank the reviewer for kind suggestion. We have corrected the typographical error in the abstract as “Vitamin K-antagonist (VKA, e.g. warfarin), commonly used.” (Page 3, line 3)

Reviewer 2

1. The author will do well to provide a justification for the following inclusion criteria: warfarin-eligible (men with CHA2DS2VASc score ≥ 1 ; women with CHA2DS2VASc score ≥ 2). Of course female gender is a known risk factor for AF-related stroke, the authors can provide a rationale for focusing on women with CHA2DS2VASc score ≥ 2 . Are the authors following a specific guideline (AHA/ESC) for stroke prevention?

Response: Thank you. We acknowledge that female gender is a risk modifier in AF-related stroke, therefore we considered stroke prevention in patients with men with CHA2DS2VASc score ≥ 1 ; and women with CHA2DS2VASc score ≥ 2 . This follows the recommendations (class IIa) from the 2020 ESC Guidelines for the diagnosis and management of atrial fibrillation and we have added the statement “Patients who are eligible for stroke prevention according to the ESC Guidelines for the diagnosis and management of atrial fibrillation (ie. men with CHA2DS2VASc score ≥ 1 ; women with CHA2DS2VASc score ≥ 2)” into the methods section, page (Page 7, line 14-17)

2. Secondly, the authors will do well to address how patients who have already being prescribed DOACs will be managed. Will they be excluded?

Response: Thank you. The patients who are switched to DOACs will be followed up for clinical events. The TTR will be analyzed until the end of warfarin treatment. We have added this statement to the text: “The patients who are switched to NOACs will be followed for clinical events. The INR data will be collected to the end of warfarin treatment for the TTR calculation and analysis as intention to treat analysis.” See methods section. (Page 15, line 15-17)